# A Network of Sites and Upskilled Therapists to Deliver Best-Practice Stroke Rehabilitation of the Arm: Protocol for a Knowledge Translation Study

**DOI:** 10.3390/healthcare11233080

**Published:** 2023-12-01

**Authors:** Leeanne M. Carey, Liana S. Cahill, Jannette M. Blennerhassett, Michael Nilsson, Natasha A. Lannin, Vincent Thijs, Susan Hillier, Dominique A. Cadilhac, Geoffrey A. Donnan, Meg E. Morris, Leonid Churilov, Marion Walker, Shanthi Ramanathan, Michael Pollack, Esther May, Geoffrey C. Cloud, Sharon McGowan, Tissa Wijeratne, Marc Budge, Fiona McKinnon, John Olver, Toni Hogg, Michael Murray, Brendon Haslam, Irene Koukoulas, Brittni Nielsen, Yvonne Mak-Yuen, Megan Turville, Cheryl Neilson, Anna Butler, Joosup Kim, Thomas A. Matyas

**Affiliations:** 1Occupational Therapy, School of Allied Health Human Services and Sport, La Trobe University, Melbourne, VIC 3086, Australia; liana.cahill@acu.edu.au (L.S.C.); natasha.lannin@monash.edu (N.A.L.); b.haslam@latrobe.edu.au (B.H.); i.koukoulas@latrobe.edu.au (I.K.); b.nielsen@cgmc.org.au (B.N.); y.mak-yuen@latrobe.edu.au (Y.M.-Y.); m.turville@latrobe.edu.au (M.T.); c.neilson@latrobe.edu.au (C.N.); anna.butler@latrobe.edu.au (A.B.); t.matyas@latrobe.edu.au (T.A.M.); 2Austin Campus, Florey Institute of Neuroscience and Mental Health, Heidelberg, VIC 3084, Australia; jannette.blennerhassett@austin.org.au (J.M.B.); vincent.thijs@florey.edu.au (V.T.); dominique.cadilhac@monash.edu (D.A.C.); joosup.kim@monash.edu (J.K.); 3School of Allied Health (Melbourne Campus), Australian Catholic University, Fitzroy, VIC 3065, Australia; 4Austin Health, Heidelberg, VIC 3084, Australia; michael.murray@austin.org.au; 5College of Health, Medicine and Wellbeing, The University of Newcastle, Callaghan, NSW 2308, Australia; michael.nilsson@newcastle.edu.au (M.N.); shanthi.ramanathan@hmri.org.au (S.R.); 6Centre for Rehab Innovations, The University of Newcastle, Callaghan, NSW 2308, Australia; michael.pollack@health.nsw.gov.au; 7Department of Neuroscience, Central Clinical School, Faculty of Medicine, Nursing and Health Sciences, Monash University, Melbourne, VIC 3800, Australia; g.cloud@alfred.org.au; 8Faculty of Medicine Dentistry and Health Sciences, The University of Melbourne, Melbourne, VIC 3010, Australia; geoffrey.donnan@unimelb.edu.au (G.A.D.); leonidc@unimelb.edu.au (L.C.); 9Allied Health and Human Performance, University of South Australia, Adelaide, SA 5001, Australia; susan.hillier@unisa.edu.au; 10Clinical Sciences at Monash Health, Faculty of Medicine, Nursing and Health Sciences, Monash University, Melbourne, VIC 3800, Australia; mbudge@bendigohealth.org.au; 11Melbourne Brain Centre, Royal Melbourne Hospital, Parkville, VIC 3050, Australia; 12Academic and Research Collaborative in Health, La Trobe University, Melbourne, VIC 3086, Australia; m.morris@latrobe.edu.au; 13The Victorian Rehabilitation Centre, Healthscope Limited, Melbourne, VIC 3150, Australia; 14School of Medicine, University of Nottingham, Nottingham NG7 2UH, UK; marion.walker@nottingham.ac.uk; 15Health Economics and Impact, Hunter Medical Research Institute, New Lambton, NSW 2305, Australia; 16Hunter New England Health, New Lambton, NSW 2305, Australia; 17UniSA Health, University of South Australia, Adelaide, SA 5001, Australia; esther.may@unisa.edu.au; 18Department of Neurology, Alfred Health, Melbourne, VIC 3004, Australia; 19National Stroke Foundation, Melbourne, VIC 3000, Australia; sharon.mcgowan@ranzcp.org; 20Department of Medicine, Melbourne Medical School, The University of Melbourne, Melbourne, VIC 3010, Australia; twi@unimelb.edu.au; 21Department of Neurology and Stroke Services, Western Health, Sunshine Hospital, Footscray, VIC 3021, Australia; 22Bendigo Health, Bendigo, VIC 3550, Australia; 23St. Vincent’s Hospital, Melbourne, VIC 3065, Australia; fiona.mckinnon@svha.org.au; 24Epworth Healthcare, Richmond, VIC 3121, Australia; john.olver@epworth.org.au; 25Barwon Health, Geelong, VIC 3220, Australia; 26Department of Occupational Therapy, Alfred Health, Melbourne, VIC 3004, Australia

**Keywords:** stroke, implementation science, neurological rehabilitation, stroke rehabilitation, somatosensory, healthcare services, occupational therapy, physiotherapy

## Abstract

Implementation of evidence-informed rehabilitation of the upper limb is variable, and outcomes for stroke survivors are often suboptimal. We established a national partnership of clinicians, survivors of stroke, researchers, healthcare organizations, and policy makers to facilitate change. The objectives of this study are to increase access to best-evidence rehabilitation of the upper limb and improve outcomes for stroke survivors. This prospective pragmatic, knowledge translation study involves four new specialist therapy centers to deliver best-evidence upper-limb sensory rehabilitation (known as SENSe therapy) for survivors of stroke in the community. A knowledge-transfer intervention will be used to upskill therapists and guide implementation. Specialist centers will deliver SENSe therapy, an effective and recommended therapy, to stroke survivors in the community. Outcomes include number of successful deliveries of SENSe therapy by credentialled therapists; improved somatosensory function for stroke survivors; improved performance in self-selected activities, arm use, and quality of life; treatment fidelity and confidence to deliver therapy; and for future implementation, expert therapist effect and cost-effectiveness. In summary, we will determine the effect of a national partnership to increase access to evidence-based upper-limb sensory rehabilitation following stroke. If effective, this knowledge-transfer intervention could be used to optimize the delivery of other complex, evidence-based rehabilitation interventions.

## 1. Introduction

Implementation of evidence-based stroke rehabilitation interventions improves patient outcomes [1,2]. Evidence-based therapies for the upper limb after stroke are recommended in clinical practice guidelines [3,4] and in international best-practice guidelines [5,6]. Yet, there is inconsistent access to and delivery of quality, evidence-based stroke rehabilitation, leading to suboptimal outcomes [3,7,8]. The need for and potential benefit of implementation interventions to promote the uptake of best-evidence rehabilitation are highlighted [9,10,11].

There are currently very few knowledge-transfer interventions of known effectiveness to facilitate practice change for complex interventions in stroke rehabilitation, and the certainty of the evidence is very low [11]. We developed an implementation intervention to drive behavior change in clinical and community settings [12]. The intervention is guided by the Theoretical Domains Framework [13,14,15], with translation strategies from the Behavior Change Wheel [16]. The intervention targets the delivery of science-based rehabilitation that requires knowledge and skill of the rehabilitation therapist, an application of knowledge translation that is virtually untested in the field of stroke rehabilitation [12].

Major evidence–practice gaps in stroke rehabilitation have been identified in addressing the loss of body sensation after stroke nationally [17] and internationally [18], contributing to poor arm use and reduced ability to return to previous life activities after stroke [19,20,21]. Impaired sensation is experienced by one in two stroke survivors [22,23,24,25]. This loss is beyond any reduction experienced with healthy aging [22,23]. As survivors of stroke report: “It is like the hand is blind” and “…I couldn’t really do daily stuff... I couldn’t hold anything, things were just dropping…so I had nothing, there was nothing there” [26]. Many learn non-use of their hand, leading to secondary problems and restricting return to valued activities and work [19,20,22,27]. In addition, upper limb sensory loss is a factor contributing to inferior results in rehabilitation outcomes [19,28,29,30], and adequate sensation is a prerequisite for full motor recovery of the paretic upper limb [31].

Despite the high prevalence and negative impact on function, it has been highlighted that loss of body sensations is a ‘neglected’ area of stroke rehabilitation [32]. Rehabilitation therapists often use a compensatory, rather than restorative, approach to somatosensory loss and recovery [17,18]. Yet, use of compensation potentially reinforces learned non-use of the limb with negative long-term consequences. In our national survey, less than half of healthcare professionals reported satisfaction with the treatments they were using, or confidence in their ability to treat somatosensory impairment after stroke, indicating a readiness to change practice [17]. Barriers to implementation of best-practice sensory rehabilitation identified by therapists include low therapist confidence; lack of skills; inconsistent access to resources; and reduced quality of therapy delivery [17,33].

The purpose of this study is to increase access to best-evidence rehabilitation of the upper limb and improve outcomes for stroke survivors. Our focus is on delivery of best-evidence somatosensory rehabilitation, given the evidence–practice gap identified nationally and internationally [17,18]. Specifically, we will use a knowledge transfer intervention to upskill therapists and deliver recommended best-evidence somatosensory rehabilitation to more survivors of stroke.

A neuroscience-based approach to rehabilitation of somatosensory impairment, known as SENSe (Study of the Effectiveness of Neurorehabilitation on Sensation) [34] (http://youtu.be/G9V3I30pn68; accessed on 2 October 2023), has been systematically developed and tested, consistent with the Medical Research Council Framework for the development and evaluation of complex interventions [35]. SENSe therapy has demonstrated efficacy across a series of studies [36,37], including a double-blinded randomized controlled trial [34], with reported improvements in somatosensory capacity [34] and performance of valued occupations [26]. Survivors of stroke report the positive experience and benefits of being involved in SENSe therapy, demonstrating acceptability of this therapy for this population [26]. SENSe therapy is recommended in clinical practice guidelines for stroke [3] and in best-practice International Standards for Arm Rehabilitation Post-stroke [5].

Skill and experience of the therapist may impact implementation and delivery of evidence-based complex interventions. This is particularly evident when service delivery requires a high level of skill from therapists [11,38]. Investigation of the impact of therapist experience on therapy outcomes is therefore warranted. Further, while evidence is growing about health outcomes from implementation interventions across various settings and health professional groups [11,39], there is a paucity of data on their cost-effectiveness to inform whether the investment of resources justifies the additional benefits that might be achieved. Therefore, this study will also investigate cost-effectiveness of the intervention.

We have created a partnership of survivors of stroke, clinicians, researchers, healthcare organizations, and policy makers to (i) increase access to evidence-based SENSe therapy delivered by therapists via a network of clinical practice settings and specialist SENSe therapy centers, and (ii) improve outcomes for survivors of stroke with somatosensory impairment of the arm/hand. The partnership is supported with a National Health and Medical Research Council Partnership grant from Australia (GNT 1134495), which has allowed us to create a centralized knowledge-translation hub and four specialist therapy centers.

Two complementary studies, SENSe Implement [12] and SENSe CONNECT (ACTRN12618001389291), are being undertaken to address the identified gap and achieve our overall aims. Together, the studies will permit investigation across different modes of delivery and skill levels of therapists and will involve approximately 100 therapists and 250 stroke survivors. The first study, SENSe Implement [12] (ACTRN12615000933550), focuses on testing the effectiveness of our knowledge-transfer intervention to change clinician behavior in existing rehabilitation services. Specifically, the aim is to determine whether evidence-based knowledge-translation strategies change the practice of occupational therapists and physiotherapists in the assessment and treatment of sensory loss of the upper limb after stroke to improve patient outcomes. This study is being conducted as a pragmatic, before–after study involving eight Australian healthcare networks and existing sub-acute and community rehabilitation services (see [12] for further details).

The second study detailed here is known as SENSe CONNECT (ACTRN12618001389291). The SENSe CONNECT study is designed to increase access to evidence-based SENSe therapy via specialist SENSe therapy centers and skilled therapists. Four new specialist SENSe therapy centers, across three states in Australia, are planned to complement and extend current services. This model of service delivery differs from the current practice model being tested in the SENSe Implement study. The focus for SENSe CONNECT is on increased access for survivors of stroke living in the community. We will create a centralized hub to lead the knowledge-translation intervention and provide upskilling of therapists. A network of specialist SENSe therapy centers and community of therapists will be linked with the hub to facilitate implementation and sustainability. Web-based resources will be developed to further support therapists and help sustain practice change. Some of the broader contextual aspects of implementation [40,41] will also be investigated in the SENSe CONNECT study, including impact of therapist expertise on outcomes and cost-effectiveness. The specific aims of the current SENSe CONNECT study are to

Increase access to evidence-based SENSe therapy delivered via a network of specialist SENSe therapy centers and skilled therapists.Improve outcomes for survivors of stroke with somatosensory impairment of the arm/hand (primary outcome—somatosensory function; secondary outcomes—performance of self-selected valued activities, arm use, and quality of life).Achieve high treatment fidelity for therapists in the delivery of upper-limb sensory rehabilitation following a tailored, evidence-informed knowledge-transfer intervention.Explore the association of the amount of therapist experience in SENSe delivery with outcomes for stroke survivors.Evaluate the cost-effectiveness of the knowledge-translation intervention in terms of the amount of improvement in SENSe therapy outcomes, i.e., somatosensory function, performance in valued activities, arm use, and quality of life.

## 2. Materials and Methods

### 2.1. Study Design

The overall knowledge translation study involves a centralized hub, 4 specialist SENSe therapy centers, and 11 healthcare networks to deliver evidence-based upper-limb sensory rehabilitation for stroke survivors. The SENSe CONNECT arm detailed here is a prospective, pragmatic knowledge-translation study, involving four specialist SENSe therapy centers and stroke survivors in the community. Inclusion criteria are broad to test broader treatment effectiveness, and therapy will be delivered using a within-participant wait list control design [42].

#### 2.1.1. Centralized Translation Hub

A centralized hub was created to lead the knowledge translation intervention and provide upskilling of therapists. This includes a multimodal approach to upskilling therapists [12], which is supported with a therapy training manual and a suite of training videos. Therapists are credentialled and treatment fidelity is monitored and supported using document audit and fidelity observations and feedback [43]. The network of specialist services and community of therapists will facilitate implementation and sustainability. Web-based resources have been developed (https://sensetherapy.net.au/; accessed on 2 October 2023) to further support therapists and help sustain practice change via a community of practice. An overview of the network of sites and centers involved is provided in Figure 1.

#### 2.1.2. Specialist SENSe Therapy Centers and SENSe CONNECT Protocol

Four specialist SENSe therapy centers have been set up to deliver SENSe therapy to stroke survivors across three states in Australia. The new specialist therapy centers are linked with healthcare networks and universities or research institutes. The SENSe therapy centers will permit delivery of therapy to stroke survivors living in the community. Therapy may be delivered at the center or in the client’s home.

Occupational therapists and physiotherapists skilled and credentialled in SENSe therapy will deliver SENSe therapy to stroke survivors across the specialist therapy centers, using a pragmatic, within-subject wait list design [42]. The wait list design will evaluate change in outcomes over a 6-week period of usual care not associated with SENSe therapy, compared to a 6-week period of SENSe therapy delivered by credentialled therapists. During the wait list phase, participants will receive usual care, which will be monitored. Upper-limb therapy will not be restricted for participants during the control wait list phase; rather, participants will receive ‘usual care’ conditions [44]. Consistent with an independent review and analysis of ‘usual care’ usually received in health services in Australia (based on national audit and knowledge-translation study), this will usually not involve specific sensory rehabilitation, such as SENSe therapy [44]. Participants may continue to receive usual care during the SENSe therapy intervention period. Our prior controlled clinical trials provide evidence that exposure to sensory stimuli alone and/or current usual care is not usually sufficient to affect clinically significant improvement in somatosensory function [34,36,37]. All services and usual care received will be monitored.

Stroke survivors attending specialist SENSe therapy centers will be assessed on four occasions: baseline (i.e., Assessment 1, A1); after 6 weeks of usual care (i.e., A2); after 6 weeks of SENSe therapy (i.e., A3); and at follow up, 12 weeks post A3 (i.e., A4). All participants attending SENSe therapy centers will receive SENSe therapy (10 sessions) over a 6-week period. Assessors for health outcomes of stroke survivors will be blinded to study design and timing of therapy delivery. Researchers and statisticians involved in the data analysis will also be blinded to study design and timing of therapy delivery.

SENSe therapists will not be blinded due to pragmatic reasons. It is planned that each therapist will deliver SENSe therapy to up to 12 stroke survivors. A secondary analysis will permit investigation of the association between stroke survivor outcomes achieved and the therapist’s amount of experience in delivering SENSe therapy. The study is approved by Austin Human Research Ethics Committee (HREC/18/Austin/153), and all experiments will be performed in accordance with the Declaration of Helsinki. Site-specific ethics approvals were also obtained. All participants will give voluntary informed consent.

Sample size: Stroke survivors—The SENSe CONNECT study design requires a within-subject, repeated measures analysis. The within-group analysis from the original SENSe randomized controlled trial (RCT) revealed a Cohen d of 0.366 for the somatosensory function outcome (repeated measures, small group *n* = 22/50; [34]). To achieve a within-group effect of d = 0.366 for a power of 0.8, a minimum sample of *n* = 61 is required. In the current study, the proposed sample size of *n* = 72 to 192 (i.e., 18 to 48 deliveries at each of the 4 centers) achieves this minimum and allows for dropout or poor recruitment. Assuming the variability in changes found in the SENSe RCT is replicated in the delivery of SENSe therapy, a maximum sample size of *n* = 192 will have a high degree of precision for estimating the effect size obtained within the SENSe therapy centers. The larger sample size will also facilitate generalizability of effect estimates to the broader target population of survivors of stroke living in the community. Therapists—At each specialist SENSe therapy center, it is planned that 3 or 4 therapists will each deliver SENSe therapy to 6 to 12 stroke survivors (i.e., 18 to 48 deliveries at each site; 72 to 192 across sites). A sample of 12 to 16 therapists was chosen to maximize replication across upskilled therapists while exploring the association between the amount of experience in delivery of SENSe therapy and therapy outcomes.

### 2.2. Participants

Participants for the SENSe CONNECT study include both therapists and stroke survivors.

Therapists: Inclusion criteria—Qualified occupational therapist or physiotherapist; current registration to practice; and willing to participate in the upskilling in SENSe therapy, undertake evaluation, and actively participate in mentoring and treatment fidelity activities to develop competency to deliver SENSe therapy. There are no additional exclusion criteria for the participants who are therapists. Replacement therapists will be recruited as needed. As a pragmatic implementation trial, restrictions will not be imposed on therapist participants’ experience or number of stroke survivors treated. However, demographic information for all therapist participants will be collected to ascertain relationships between stroke survivor outcomes and therapist characteristics.

Stroke survivors: People with stroke living in the community and presenting with new or chronic somatosensory impairment will be recruited. Stroke survivors may be referred via partner organizations, neurologists, rehabilitation physicians, general practitioners, other organizations, or self-referral. Inclusion criteria: A clinical diagnosis of stroke (including infarct or hemorrhage, people with a second stroke, no restriction on time since stroke); impaired touch sensation, limb position sense, and/or tactile object recognition of the upper limb, identified clinically and with screening tests; medically stable; able to give informed consent; able to comprehend simple instructions; willing to commit time to participate in the SENSe therapy program; and living in the community. Key exclusion criteria: Sensory impairment not due to stroke; severe unilateral spatial neglect (measured via line bisection and shape cancellation task); prior history of other central nervous system dysfunction with an unstable or progressive prognosis; severe to moderate cognitive impairment (i.e., not able to comprehend simple instructions or sustain attention needed to participate in treatment); not able to give informed consent; physical limitations that prevent participation in therapy tasks (e.g., contracture of the hand, or unhealed wounds); and unable to participate in a clinical appointment lasting 30 min. Thus, the design tests effectiveness with relatively unselected survivors of stroke in the community.

### 2.3. Setting

Specialist SENSe therapy centers will be based at four metropolitan sites in Australia, each having a link with a health setting and an academic or research organization. Sites are Austin Health (link with Florey Institute of Neuroscience and Mental Health and La Trobe University) and Alfred Health (link with Monash University) in Melbourne, Victoria; John Hunter Hospital (link with University of Newcastle) in Newcastle, NSW; and UniSA Health (link with University of South Australia) in Adelaide, SA.

### 2.4. Outcomes

Access to evidence-based sensory rehabilitation: The primary outcome is the number of complete deliveries of SENSe therapy at the SENSe therapy center by therapists credentialled in SENSe therapy. A delivery will be counted as complete if it meets the criterion of at least 7 sessions delivered with a stroke survivor.

Stroke survivors: The primary outcome is change in arm somatosensory function, pre–post SENSe therapy, across three somatosensory domains, measured using standardized, quantitative somatosensory measures and a normalized summary impairment index calibrated to normed age-matched performance [34,45]. The index will be derived from scores on the Tactile Discrimination Test [46], Wrist Position Sense Test [47], and functional Tactile Object Recognition Test [48]. Each of these quantitative measures assesses the person’s ability to discriminate different somatosensations; i.e., texture discrimination, wrist joint proprioception, and haptic object recognition of the upper limb. Comparable ranges of impairment from just noticeable to extreme impairment defined for each measure enable the normalization of the three test scales for comparison in clinical and research settings [45].

The following secondary outcomes will also be assessed: client-rated performance and satisfaction of valued activities, using the Canadian Occupational Performance Measure (COPM) [49] and clinician-rated performance on the same valued activities using the Performance Quality Rating Scale for Somatosensation after Stroke [50]; arm use, using Motor Activity Log-14 [51]; and health-related quality of life, using the Assessment of Quality of Life (AQoL-6D) [52]. See Table 1 for the schedule of study outcomes.

Therapists: Treatment fidelity is a primary outcome for therapists and will be assessed as the ability to deliver SENSe therapy with high fidelity measured with a criterion-based checklist (comprising 29 components core to the delivery of SENSe therapy; 80% fidelity) and customized documentation audit checklist [43]. Fidelity will be assessed by an independent person, using a document audit, for each therapist after delivery of each 6-week program of SENSe therapy. In addition, a sample of treatment sessions (early, middle, and late in the sequence of 10 sessions for the stroke participant) will be observed and rated by an independent person using a session-based treatment fidelity checklist. Therapist participants will also be assessed for change in practice behaviors related to SENSe therapy (secondary outcome), specifically change in knowledge, skill, and confidence levels. These outcomes will be assessed using pre–post implementation questionnaires [12] adapted for the SENSe CONNECT study. The pre-questionnaire will be completed prior to upskilling, credentialing, and first delivery of SENSe therapy. The post-implementation questionnaire will be completed after delivery of therapy to the 12th stroke survivor treated, or following the last scheduled delivery of therapy, for that therapist.

Therapist experience effect: Therapist experience will be measured according to the number of completed deliveries of the SENSe therapy program, with delivery of 7 or more sessions per stroke survivor the criterion for a completed delivery. Each therapist is anticipated to deliver SENSe therapy to 6 to 12 survivors of stroke.

Resource utilization and therapy costs: Resources used by survivors of stroke will be collected at all assessment occasions for the usual care wait list and intervention periods using a patient/carer survey. Resources will be assigned prices to convert these into costs using contemporaneous Australian reference sources. Costs will be inflated/deflated for a common reference year (e.g., 2022) using the Total Health Price Index [53], as required. Program level costs of intervention delivery: SENSe therapy costs will include the cost of upskilling therapists, equipment, and delivering the intervention according to the model of care established, as well as determining the cost of usual care. The type and quantity of each resource used will be collected in natural units, e.g., number of sessions and time taken to train therapists.

### 2.5. Implementation Intervention

The current knowledge-translation study and implementation intervention developed by the research team are based on the Theoretical Domains Framework [14], and guided by strategies from the Behaviour Change Wheel [16] to facilitate practice change and Normalisation Process Theory [54] to enhance sustainability. Specifically, the multi-component implementation intervention to support knowledge translation includes (i) interactive training workshops, (ii) establishing a clinical lead and site champions; (iii) provision of educational materials and structured therapy booklets; and (iv) use of treatment fidelity checklists [43] to guide feedback on therapy and to assess outcomes.

Upskilling of therapists: Occupational therapists and physiotherapists will be upskilled in delivery of SENSe therapy by a trained clinical lead researcher from the central hub using multimedia resources. Training includes a 1.5-day interactive workshop, involving theory (1 h), practical hands-on SENSe training sessions (7 h), and applied treatment planning (2 h), and is supported by 3 independent learning modules that include supervised practice tasks and case scenarios (estimated to take 1 to 2 h each); the total time is approximately 13 to 16 h. Therapists are credentialed, via observation of a simulated therapy session at the end of the upskilling process. Participant therapists will also receive supervision and mentoring during delivery of SENSe therapy. This will be primarily via the treatment fidelity observations and audit feedback. Therapists are introduced to the treatment fidelity checklist [43] during upskilling sessions and encouraged to use it for self-evaluation. The treatment fidelity criterion checklist is also used by SENSe therapy trainers to provide feedback to therapists on treatment notes and observed therapy delivery.

### 2.6. SENSe Therapy Intervention

SENSe (Study of the Effectiveness of Neurorehabilitation on Sensation; [34]), is a science- and evidence-based rehabilitation therapy designed to help people with stroke regain a sense of touch and use it in daily activities (http://youtu.be/G9V3I30pn68; accessed on 2 October 2023). As such, the focus is on restoration of function rather than compensation. Clinical practice protocols and therapy tools [55] have been produced to facilitate implementation and quality delivery in clinical settings. The intervention will be delivered face to face by skilled and credentialled occupational therapists and/or physiotherapists within specialist SENSe therapy centers. SENSe therapy involves modules of sensory discrimination training of texture discrimination, sensing the position of the upper limb in space (proprioception), recognizing objects through the sense of touch, and learning how to apply these skills in daily tasks identified by the person with stroke (Figure 2). Seven training principles are used during the process of somatosensory discrimination training as follows: select; attentive exploration; feedback; calibrate; anticipate; repeat and progress; and transfer (http://youtu.be/G9V3I30pn68; accessed on 2 October 2023). Specially designed training tools are used and include grids and texture training wheels of surfaces with varying surface features, graded for large, medium, and fine surface differences; box-like apparatus and protractor scales for training wrist position sense; and graded sets of objects for functional tactile object recognition that train discrimination and recognition of object weight, crushability, shape, size, temperature, texture, and functional motion. The client selects two valued activities that they believe are impacted by their sensory loss to focus on in therapy. They are guided to discover the sensory challenges in the activity and how they can use their new skills to perform the task better. Examples include using a knife or fork, finding money in a wallet, doing up buttons, and using a remote-control device. The intervention will be tailored according to the level of impairment and functional goals of the stroke participant. The dose is 10 1-hour sessions, over a period of 6 weeks. The frequency of sessions is approximately twice a week. The number, duration, and specific content of all sessions are monitored using customized training forms.

### 2.7. Methods to Facilitate Sustainability

A community of practice (CoP) will be developed to support therapists in the implementation of evidence-based SENSe therapy. Initially, this community will be developed locally as part of the peer group upskilling of therapists at SENSe therapy centers. A website and online presence will be developed to enhance this CoP and provide ongoing education and peer support in the sustained implementation of evidence-based SENSe therapy. The website will connect therapists, provide support and information with case scenarios, and permit interactive feedback.

### 2.8. Data Analysis

Data management and data statement: Due to the personal nature of the data and original ethics approval, the data will not be made available broadly. De-identified data may be made available for related research and analyses by the research group and collaborators with additional ethics approval. Data will be entered using the REDCap electronic data capture tools [56], hosted at the Florey Institute of Neuroscience and Mental Health, by the site coordinator or trained delegate. REDCap features are in place to ensure valid data capture (e.g., data range checks), and data quality checking procedures are in place to ensure accuracy at initial entry and subsequently checked by a second independent researcher. Data will be de-identified for analyses. All data will be kept secure and confidential as per the approved ethics protocol.

Access to evidence-based sensory rehabilitation: The number of stroke survivors who receive SENSe therapy by therapists credentialled in SENSe therapy will be calculated. Successful completion of SENSe therapy for a survivor of stroke is determined as delivery of 7 or more therapy sessions. The number of attempted deliveries of SENSe therapy that did not meet the criterion for successful completion will also be counted.

Effect of SENSe therapy: We will evaluate the therapeutic gain in arm somatosensory function for stroke survivors (primary outcome) during the SENSe therapy phase compared to that of the usual care phase by estimating the mean within-person difference between the two trends and the associated 95% confidence intervals. The magnitude of change in arm somatosensory function under the two conditions will be calculated as a standardized effect size and compared to the benchmark effect size obtained in the original SENSe RCT (comparable cohort) [34]. Statistical estimates of change will be reported with 95% confidence intervals, and the extent of overlap across studies in these estimates of change will be described. Confidence intervals for the difference between mean change scores from the present and benchmarked study [34] will also be obtained. The analysis design will include the SENSe therapy center as a variable, to evaluate its potential systematic effect. Individual participant characteristics that may impact magnitude of therapeutic gain, such as age and time post-stroke, will be explored. Improvement in secondary outcomes, i.e., clinician-rated performance in valued activities, arm use, and quality of life, will also be evaluated pre–post therapy. Depending on verification of the normal distribution assumption, either a t-distribution or a bootstrapping method will be used to obtain the confidence interval. Maintenance of the intervention effect will be evaluated at the 12-week follow up (A4).

Treatment fidelity and behavior change: High fidelity delivery will be defined as a score of 80% or higher on the SENSe treatment fidelity checklists [43]. Audit data from treatment sessions will be independently summarized for each SENSe therapy treatment program delivered. The number of therapy deliveries achieving high treatment fidelity will be summarized. Pre- and post-implementation questionnaires of therapist knowledge, confidence, and ability to deliver SENSe therapy will be summarized and response tendencies will be examined using contingency tables, McNemar’s test, and graphical representation.

Therapist experience effect: The availability of 6 to 12 sequential cases treated with SENSe therapy by each of the 12 to 16 therapists will allow exploration of the hypothesis that a trend towards better outcomes arises with experience of therapy delivery. Individual therapeutic gain scores will be adjusted for initial sensory impairment, consistent with our prior analysis of which individual variables affect therapeutic gain [57]. Growth curve models will be explored for the sequence of adjusted therapeutic gains from each of the therapists (*n* = 12–16) to evaluate the autocorrelation structure of residuals and the viability of a common model. A pooled estimate of the effect of treatment experience will then be obtained via a meta-analysis, or hierarchical linear modelling if a common trajectory form is applicable.

Economic evaluation: We will report a cost description analysis for usual care and the SENSe therapy interventions and summarize these data as part of a cost consequence analysis [58] presenting the disaggregated costs and primary and secondary outcomes for stroke participants. The incremental cost-effectiveness of SENSe therapy compared to monitored wait list usual care will also be evaluated as the cost per achievement of improved arm somatosensory function (primary outcome for stroke survivor). We will also use simulation modelling to estimate the incremental cost per Quality Adjusted Life Year gained with SENSe therapy, drawing data from the study and the published RCT for SENSe [34]. One-way sensitivity and multivariable uncertainty analyses will be performed to assess the robustness of results. The findings from the economic evaluation will be used to inform business cases for future adoption of SENSe within the Australian healthcare context and may have relevance to other countries with similar funding models for healthcare. Results will be reported according to the Consolidated Health Economic Evaluation Reporting Standards (CHEERS) [59,60].

### 2.9. Research Impact Evaluation

The potential translational impact of the research will be evaluated using the Framework for the Assessment of Impact and Translation (FAIT) [61]. A program logic model that defines aims, activities, outputs, stakeholders, and impact will guide this process. Impact will be evaluated in relation to the following domains: advance knowledge, e.g., publication metrics, presentations, and social media; capacity and capability building, including partnerships and networks, formal education training, upskilling of therapists, and integration with policy/practice; implementation, in relation to clinical practice change and research practice change; community benefit, including access to services and health outcomes for stroke survivors; economic benefits, such as employment-created, cost-effectiveness of SENSe therapy, commercialization, grants leveraged, potential downstream savings, and increase in potential lifetime earnings; and policy and legislation, including policy representation, policy relationships built and direct translation to policy.

### 2.10. Patient and Public Involvement

People who have experienced a stroke have identified the practice gap and helped drive this research from its inception. The Stroke Foundation, an Australian public advocacy group, is a partner in this program of research. Survivors of stroke and Stroke Foundation representatives are named on the successful Partnership grant and involved in group meetings. They have engaged in discussions and formulation of the research questions, design, and conduct of the study, choice of outcome measures, and recruitment to the study both at the initial project planning stage and during team meetings. They have also contributed to plans for dissemination and impact. Involvement is consistent with Stroke Foundation guidelines for involvement of people with lived experience in research.

## 3. Discussion

Access to best-evidence rehabilitation therapies after stroke continues to be an issue in Australia [3] and globally [7,62,63,64]. To address this issue, and to demonstrate a much-needed approach for stroke rehabilitation knowledge translation, we created a model based on a collaborative partnership approach, to meet the specific needs of stroke rehabilitation stakeholders, and to complement existing services. The centralized hub and specialist therapy centers created are designed to provide a vehicle to increase access to best-practice therapy through connecting all key stakeholders: stroke survivors, clinicians, healthcare organizations, research institutes, universities, policy makers, funders, and the Stroke Foundation. Thus, our approach not only addresses change at the ‘micro’ level of health (i.e., individual therapists) but also at the organizational or macro-system level of healthcare, with the formation of our partnership and creation of a translation hub and specialist therapy centers. It is anticipated that sustainability will be enhanced by the structure that links the central hub with local sites and skilled, credentialled therapists. Involvement of all stakeholders will help maximize meaningful and sustained policy and practice change.

We chose to focus on the evidence–practice gap of sensory rehabilitation after stroke, identified nationally [17] and internationally [18]. Consistent with recommendations for moving research evidence into practice [65], we sought to implement SENSe therapy as this therapy makes explicit the details of the intervention with therapy protocols [55], and has guidelines to assess for treatment fidelity [43]. SENSe therapy is underpinned by strong evidence including a double-blind RCT [34]. A pooled analysis of individual differences indicates adult survivors of stroke with varying age, lesion of left or right hemisphere, severity of impairment, cognition, and varying time post-stroke can benefit from SENSe therapy [57]. Despite this therapy being recommended in international best-practice guidelines for rehabilitation of the upper limb [5] and national stroke guidelines [3], adoption in clinical practice has been slow.

The knowledge-translation approach and implementation intervention outlined in this protocol may provide a foundation for creating a template for knowledge translation of evidence-based stroke rehabilitation, optimized for application to therapies that are predicated on skilled delivery by therapists, as is the case for SENSe therapy. While it is acknowledged that this implementation intervention addresses only one unmet need of stroke survivors (when many have multiple deficits and needs), the knowledge translation approach has potential for broader application in specialist upskilling and creation of networks of sites to support specialized, evidence-based therapeutic interventions.

A pragmatic approach was selected to evaluate the effectiveness of our implementation intervention in real-world clinical practice and community settings, and across a range of therapists, to maximize generalizability of the results [66]. Further, we will have the opportunity to explore a therapist experience effect on SENSe outcomes. Cost-effectiveness data may be used to improve resource allocation in different service settings. It is anticipated that the costs of services provided by such centers and services could be covered by national health and disability schemes and/or private health insurers. Dissemination will be enhanced via a knowledge translation hub, specialist therapy centers, websites (e.g., https://sensetherapy.net.au/; accessed on 2 October 2023), and a community of skilled therapists embedded in a range of healthcare settings.

Limitations: Increased access to delivery of evidence-based SENSe therapy via specialist SENSe therapy centers, the primary outcome, will likely be impacted by restrictions imposed during the COVID-19 pandemic. The ability to administer the protocol as planned and deliver SENSe therapy without interruption will also likely be impacted. Changed service demands associated with the pandemic may also influence recruitment and retention of therapists and delivery of usual care. Any protocol variations and impacts of the pandemic will be monitored and reported. As a pragmatic trial, factors such as individual characteristics of treating therapists (e.g., prior expertise, number of years of practice, number of stroke survivors treated before upskilling) and survivors of stroke (e.g., age, time since stroke, concomitant impairments) will not be controlled for. These factors will, however, be monitored and explored for their potential influence on outcomes in our planned analyses. Therapists who deliver usual care will be different from those who deliver SENSe therapy. It is recognized that this may impact outcomes. It is noted, however, that involvement of different therapists in this pragmatic design is consistent with variation in therapist experience typically experienced. Further, it is not possible for the SENSe therapists to deliver usual care following upskilling, as the specialist training may bias their usual care approach. Usual care will not be restricted, nor will it involve standard protocols. Rather, there will be monitoring and comparison to usual care as defined in our aligned SENSe Implement study, which included a phase of delivery of usual care (*n* = 86 patients) before therapists were upskilled in existing healthcare services, and with an independent analysis of the care usually received in health services based on a national audit [44]. To date, there are no results to present for the current prospective study.

## 4. Conclusions

The potential impact of this study lies in bringing together producers and users of knowledge as partners to create a network of sites and ‘upskilled’ therapists to deliver best-practice stroke rehabilitation. It is hoped that through linking the evidence, therapists, and stroke survivors, this interconnected network will enable increased access to evidence-based upper-limb therapy in stroke rehabilitation and better outcomes for people who experience impaired somatosensation after stroke. If successful, there is potential for this approach to be transferred to other specialized evidence-based rehabilitation interventions.

## Figures and Tables

**Figure 1 healthcare-11-03080-f001:**
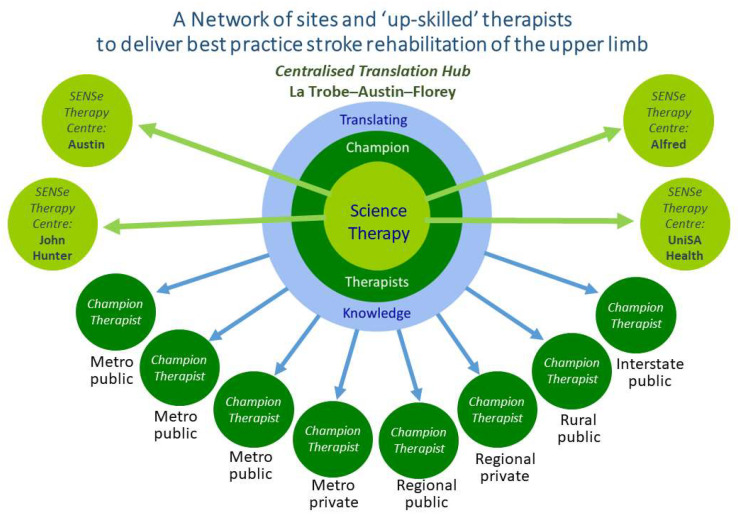
Diagram of the centralized translation hub and delivery sites, with champion therapists at existing health settings (SENSe Implement study) and specialist therapy centers (SENSe CONNECT study). Each interact with and are supported by the central hub.

**Figure 2 healthcare-11-03080-f002:**
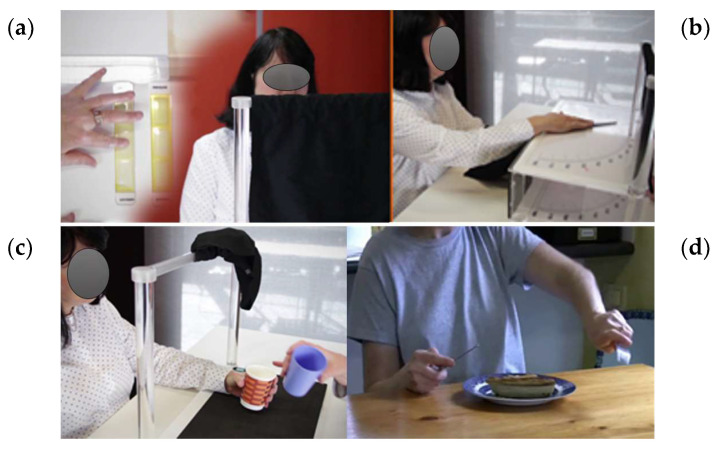
Images of SENSe therapy specialized equipment and activities used in training modules, including (**a**) discrimination of texture grids; (**b**) training of wrist proprioception using the box-like apparatus and protractor scales; (**c**) training of graded sets of everyday objects that vary in objects with varying diagnostic attributes such as crushability, shape, size, and weight; and (**d**) training in the context of self-chosen valued activities impacted by sensory loss.

**Table 1 healthcare-11-03080-t001:** Study Schedule of Assessments: SENSe CONNECT.

Outcome	Measure	A1 (Baseline)	A2 (6 Weeks Post A1)	A3 (6 Weeks Post A2)	A4 (12 Weeks Post A3)
*Survivor of Stroke*					
Arm somatosensory function	Tactile Discrimination Test (TDT)	X	X	X	X
	Wrist Position Sense Test (WPST)	X	X	X	X
	Functional Tactile Object Recognition Test (fTORT)	X	X	X	X
Performance of valued activities	Canadian Occupational Performance Measure (COPM) Performance Quality Rating Scale for Somatosensation after Stroke	X X	X X	X X	X X
Arm use	Motor Activity Log (MAL)-14 item version	X	X	X	X
Health-related quality of life	Australian Quality of Life (AQoL-6D)	X	X	X	X
Resource utilization	Resource Use and Productivity Questionnaire	X	X	X	X
Other	National Institute of Health Stroke Scale (NIHSS)	X			
	Modified Rankin Scale (mRS)	X			
	Montreal Cognitive Assessment (MoCA)	X			
	Jebsen Taylor Hand Function Test (JTHFT)	X			
*SENSe Therapist*					
Treatment fidelity	Customized Documentation Audit Checklist	Post delivery of SENSe therapy to each survivor of stroke.			
Practice behavior change	Pre–Post Implementation Questionnaires	Prior to first delivery of SENSe therapy and after delivery of therapy to 12th survivor of stroke, or last scheduled delivery for that therapist.			

## Data Availability

The datasets generated and/or analyzed during the current study are not publicly available due to the personal nature of the data and original ethics approval, but may be available from the corresponding author on reasonable request.

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
