# Peer review of "A Network of Sites and Upskilled Therapists to Deliver Best-Practice Stroke Rehabilitation of the Arm: Protocol for a Knowledge Translation Study"

_healthcare, 2023, doi:10.3390/healthcare11233080_

Round 1

Reviewer 1 Report

Comments and Suggestions for Authors

Please, write clearly your purpose of the study in a paragraph. 

Please, write in a section details about SENSe.

Author Response

Thank you. Please find responses below. The final revised and tracked manuscript is also attached. 

Comment 1: Please, write clearly your purpose of the study in a paragraph. 

Thank you. We have now added the purpose of the study explicitly in a paragraph the introduction, as follows (lines 115-122):

The purpose of this study is to increase access to best-evidence rehabilitation of the upper-limb and improve outcomes for stroke survivors. Our focus is on delivery of best-evidence somatosensory rehabilitation, given the evidence-practice gap identified nationally and internationally [17, 18]. Specifically, we will use a knowledge transfer intervention to upskill therapists and deliver recommended best-evidence somatosensory rehabilitation to more survivors of stroke. Somatosensory impairment and recovery will be quantified using well validated measures with age-matched normative standards and strong psychometric and neuroimaging foundations [34-38].

Please also note that, as in the original manuscript, the objectives are stated in the abstract (lines 58-59) and specific aims are listed at the end of the introduction from line 184.

Comment 2: Please, write in a section details about SENSe.

Thank you. Background information on SENSe are included in the Introduction in lines 123 to 133. We have now added extra details about SENSe in the Methods, lines 436 to 447, as below (in red).

SENSe therapy involves modules of sensory discrimination training of texture discrimination, sensing the position of the upper limb in space (proprioception), recognizing objects through the sense of touch, and learning how to apply these skills in daily tasks identified by the person with stroke. Seven training principles are used during the process of somatosensory discrimination training as follows: select; attentive exploration: feedback; calibrate; anticipate; repeat and progress; transfer (http://youtu.be/G9V3I30pn68). Specially designed training tools used include: grids and texture training wheels of surfaces with varying surface features, graded for large, medium and fine surface differences; box-like apparatus and protractor scales for training wrist position sense; and graded sets of objects for functional tactile object recognition that train discrimination and recognition of object weight, crushability, shape, size, temperature, texture, and functional motion. The client selects two valued activities that they believe are impacted by their sensory loss to focus on in therapy. They are guided to discover the sensory challenges in the activity and how they can use their new skills to perform the task better. Examples include, using a knife or fork, finding money in wallet, doing up buttons, using a remote-control device. The intervention will be tailored according to level of impairment and functional goals of the stroke participant. The dose is 10 one-hour sessions, over a period of 6 weeks. The frequency of sessions is approximately twice a week. The number, duration and specific content of all sessions is monitored using customized training forms.

Reviewer 2 Report

Comments and Suggestions for Authors

First, when introducing stroke, it is better to add some recent works about stroke rehabilitation, such as "Development of an untethered adaptive thumb exoskeleton for delicate rehabilitation assistance, IEEE Transactions on Robotics 38 (6), 3514-3529".

Second, the results of the paper should be stated more clearly to show the contribution of the paper. It is better to add some quantitative outcomes to the paper.

Comments on the Quality of English Language

Minor editing of English language required

Author Response

Thank you. Please find responses below. The final revised and tracked manuscript is also attached.

Comment 1: First, when introducing stroke, it is better to add some recent works about stroke rehabilitation, such as "Development of an untethered adaptive thumb exoskeleton for delicate rehabilitation assistance, IEEE Transactions on Robotics 38 (6), 3514-3529".

We thank the reviewer for highlighting the importance of using recent works about stroke rehabilitation and providing an example. Given the focus on implementation of best-evidence stroke rehabilitation we have focused on evidence practice guidelines, as per the text below. As you will note, 3 of the 4 references included are from 2022 and 2023. The app referred to in reference 5 is regularly updated and so includes works about recent stroke rehabilitation that meet the criteria of the app guidelines.

Evidence-based therapies for the upper limb after stroke are recommended in clinical practice guidelines [3, 4] and in international best-practice guidelines [5, 6].

  1. Stroke Foundation, Clinical guidelines for stroke management 2022 (living guidelines). 2022: Stroke Foundation https://informme.org.au/en/Guidelines/Clinical-Guidelines-for-Stroke-Management.
  2. Intercollegiate Stroke Working Party. National clinical guideline for stroke for the united kingdom and ireland. 2023, Royal College of Physicians: London, UK DOI: https://www.strokeguideline.org/contents/.
  3. Wolf, S.L.; Kwakkel, G.; Bayley, M.; McDonnell, M.N.; for the Upper Extremity Stroke Algorithm Working Group. Best practice for arm recovery post stroke: An international application. Physiotherapy, 2016. 102(1), 1-4 DOI: http://dx.doi.org/10.1016/j.physio.2015.08.007.
  4. Mead, G.E.; Sposato, L.A.; Sampaio Silva, G.; Yperzeele, L.; Wu, S.; Kutlubaev, M.; Cheyne, J.; Wahab, K.; Urrutia, V.C.; Sharma, V.K.; Sylaja, P.N.; Hill, K.; Steiner, T.; Liebeskind, D.S.; Rabinstein, A.A. A systematic review and synthesis of global stroke guidelines on behalf of the world stroke organization. Int J Stroke, 2023. 18(5), 499-531 DOI: 10.1177/17474930231156753.

Comment 2: Second, the results of the paper should be stated more clearly to show the contribution of the paper. It is better to add some quantitative outcomes to the paper.

Thank you. As this paper is a protocol paper, we do not have quantitative results to add at this time. Such results will be made available for the paper that reports on the findings. Please note that quantitative outcome measures are included in the design and will be reported when results are available.

Comments on the Quality of English Language

Comment 3: Minor editing of English language required

Please note that we have closely reviewed the texted and included minor editing as suggested. These changes are highlighted in the revised manuscript using track changes.

Reviewer 3 Report

Comments and Suggestions for Authors

The manuscript by Carey et al outlines the protocol aimed at increase access to evidence-based upper-limb rehabilitation for stroke survivors and improve sensorimotor functional outcomes among them. In my view, the protocol, in general, could be of interest to the broader audience spanning public health, rehabilitation outcomes, and public policy. I surely think the manuscript has merit for publication if some of the major concerns (highlighted below) could be addressed:

1.  As currently describe, it is unclear how SENSe CONNECT adds to the objectives that will be (potentially) satisfied by the SENSe Implement protocol. Most of the ‘specific aims’ of SENSe CONNECT seem essential to conduct SENSe Implement protocol thereby, challenging the notion of two protocols being complementary to each other. Perhaps the authors could consider unifying the two protocols.

2. The control group for evaluating the effectiveness of SENSe therapy is noted to be the one in which stroke survivors “may continue to receive usual care during the SENSe therapy intervention period; however, this will usually not involve therapy for the upper limb nor sensory rehabilitation, and all services and usual care received will be monitored” (lines 224-226). Not letting the control group of stroke survivors undergo upper limb rehabilitation during the SENSe therapy session restricts the scientific evaluation of therapists’ delivery of SENSe therapy vs. ‘usual care’. One of the important aspect of the current protocol would be to let the same therapists deliver two therapies (SENSe and ‘usual care’) to delineate whether their ‘upskilling’ in SENSe therapy is attributable to any improvements potentially observed from the study over those via the upper limb rehabilitation that stroke patients usually receive (‘usual care’ in principle). Consequently, the control group of stroke survivors must be exposed to adequate upper limb rehabilitation to ensure scientific merit of the study. Moreover, the therapists delivering this ‘usual care’ protocol to stroke survivors should be ensured to follow the standard protocols and checkpoints need to be at place to ensure no bias among the therapists to bolster the effectiveness of SENSe over ‘usual care’. The manuscript should also clearly describe the set of procedures involved in the usual care protocols. Notably, this step is essential to associate improvement in stroke survivors’ upper limb sensorimotor function to therapists’ delivery of SENSe therapy over ‘usual care’.

3. One of the scientific limitations with restoring upper limb movement in stroke is the lack of comparative physiological models and upper limb tasks that delineate loss of functions beyond that known with healthy aging. This is especially critical as there is evidence of nuanced deficits in upper limb sensorimotor functions with normal aging, beyond the conventional gross motor deficits attributable to muscle dynamics. Consequently, the manuscript needs to highlight the loss of sensorimotor function post stroke that is above and beyond that documented in healthy older individuals. Following articles could help authors contextualize this point in the introduction section:

a. Pagano S, Fait E, Brignani D, Mazza V. Object individuation and compensation in healthy aging. Neurobiol Aging. 2016 Apr;40:145-154. doi: 10.1016/j.neurobiolaging.2016.01.013

b. Rao N, Mehta N, Patel P, Parikh PJ. Effects of aging on conditional visuomotor learning for grasping and lifting eccentrically weighted objects. J Appl Physiol (1985). 2021 Sep 1;131(3):937-948. doi: 10.1152/japplphysiol.00932.2020

c. Ingram LA, Butler AA, Lord SR, Gandevia SC. Use of a physiological profile to document upper limb motor impairment in ageing and in neurological conditions. J Physiol. 2023 Jun;601(12):2251-2262. doi: 10.1113/JP283703

4. While the authors state that therapists will not be blinded towards the delivery of SENSe therapy to stroke survivors for ‘pragmatic reasons’, it is essential to ensure that the researcher and/or the personnel conducting data processing and analysis of the primary and secondary outcomes is blinded to the assignment of stroke survivors to either group, to reduce the chance of implicit biases.

5. Inclusion/exclusion criteria for therapists to be enrolled in the SENSe therapy training need to be standardized to ensure factors including (but not limited to) prior expertise, number of years of practice, number of stroke survivors treated before enrollment to SENSe ‘upskilling’, etc. does not structurally affect the outcomes of the study.

6. It is important to highlight whether the time since stroke will be incorporated as an inclusion criterion for recruitment of stroke survivors. Homogenizing this factor is important because therapists delivering SENSe therapy to stroke survivors within their first 6 months post stroke could observe a greater recovery of sensorimotor function to that observed among survivors having past 12 months or more post stroke. Not restricting this factor could potentially confound the effect and interpretation of SENSe therapy on the outcomes.

7. Why is the Fugl-Myer Upper Extremity test (FM-UE, an established test to evaluate the state of upper extremity function in stroke survivors) not included in one of the outcomes of this protocols?

8. Assignment (random or otherwise) of the stroke survivors to SENSe therapies/'usual core' groups needs to be described in the manuscript. 

Comments on the Quality of English Language

Proof-reading could help fix occasional typos in the text.

Author Response

Thank you. Please find responses below, with added text in red. The final revised and tracked manuscript is also attached.

The manuscript by Carey et al outlines the protocol aimed at increase access to evidence-based upper-limb rehabilitation for stroke survivors and improve sensorimotor functional outcomes among them. In my view, the protocol, in general, could be of interest to the broader audience spanning public health, rehabilitation outcomes, and public policy. I surely think the manuscript has merit for publication if some of the major concerns (highlighted below) could be addressed:

Thank you. We appreciate the positive comments and have addressed the issues below. We believe that addressing these issues have improved the manuscript.

  1. As currently describe, it is unclear how SENSe CONNECT adds to the objectives that will be (potentially) satisfied by the SENSe Implement protocol. Most of the ‘specific aims’ of SENSe CONNECT seem essential to conduct SENSe Implement protocol thereby, challenging the notion of two protocols being complementary to each other. Perhaps the authors could consider unifying the two protocols.

The SENSe Implement protocol is already published (Cahill et al., 2018). That protocol focuses on behaviour change of clinicians in existing rehabilitation settings. By comparison, the SENSe CONNECT protocol is focused on development of specialist SENSe therapy centres and a translation hub to increase access to SENSe therapy, while recognising and complementing existing services. SENSe Implement does not use the same partnership of individuals and stakeholders as for SENSe CONNECT.

In relation to the specific aims of SENSe CONNECT, please find associated comments below (in italics).

The specific aims of the current SENSe CONNECT study are to:

  • Increase access to evidence-based SENSe therapy delivered via a network of specialist SENSe therapy centers and skilled therapists - Increased access via specialist SENSe therapy centers is specific to the SENSe CONNECT study. It is a primary outcome for SENSe CONNECT.  
  • Improve outcomes for survivors of stroke with somatosensory impairment of the arm/hand (primary outcome - somatosensory function; secondary outcomes - performance of self-selected valued activities, arm use, and quality of life) – The primary outcome is relevant to SENSe CONNECT and SENSe Implement, but will be analysed separately given the different context. SENSe CONNECT involves some additional secondary outcomes.
  • Achieve high treatment fidelity for therapists in the delivery of upper limb sensory rehabilitation following a tailored, evidence-informed knowledge transfer intervention. – This outcome is relevant to both studies. However, the context will vary for SENSe CONNECT where the focus is on training expert therapists and providing additional individual therapist level feedback on treatment fidelity during the process of upskilling and delivery.
  • Explore the association of amount of therapist experience in SENSe delivery on outcomes for stroke survivors. This aim is specific to SENSe CONNECT and will be mapped using trend analysis across the 12 deliveries to explore the therapist experience effect.
  • Evaluate the cost-effectiveness of the knowledge translation intervention relative to unit of improvement in SENSe therapy outcomes, i.e. somatosensory function, performance in valued activities, arm use, and quality of life. – This analysis is specific to SENSe CONNECT, given data collected for each client in this study (such data is not collected in the SENSe Implementation study).
  1. The control group for evaluating the effectiveness of SENSe therapy is noted to be the one in which stroke survivors “may continue to receive usual care during the SENSe therapy intervention period; however, this will usually not involve therapy for the upper limb nor sensory rehabilitation, and all services and usual care received will be monitored” (lines 224-226). Not letting the control group of stroke survivors undergo upper limb rehabilitation during the SENSe therapy session restricts the scientific evaluation of therapists’ delivery of SENSe therapy vs. ‘usual care’. One of the important aspect of the current protocol would be to let the same therapists deliver two therapies (SENSe and ‘usual care’) to delineate whether their ‘upskilling’ in SENSe therapy is attributable to any improvements potentially observed from the study over those via the upper limb rehabilitation that stroke patients usually receive (‘usual care’ in principle). Consequently, the control group of stroke survivors must be exposed to adequate upper limb rehabilitation to ensure scientific merit of the study. Moreover, the therapists delivering this ‘usual care’ protocol to stroke survivors should be ensured to follow the standard protocols and checkpoints need to be at place to ensure no bias among the therapists to bolster the effectiveness of SENSe over ‘usual care’. The manuscript should also clearly describe the set of procedures involved in the usual care protocols. Notably, this step is essential to associate improvement in stroke survivors’ upper limb sensorimotor function to therapists’ delivery of SENSe therapy over ‘usual care’.

Upper limb therapy will not be restricted during the control wait list period, it will be ‘usual’ care conditions. Comments made in relation to ‘usual care’ are based on an independent analysis of the care usually received in health services based on a national audit and knowledge translation study in Australia. This review and analysis is reported in the paper by Cahill et al (Cahill et al., 2022) and covers usual care for upper limb therapy after stroke. We have now linked our previous text more explicitly with the independent evidence of usual care in Australia, and make reference to the paper in which it is reported, as below (in red). Further, evidence from our ‘… prior controlled clinical trials provide evidence that exposure to sensory stimuli alone and/or current usual care is not usually sufficient to effect clinically significant improvement in somatosensory function [39, 41, 42].’

During the wait list phase participants will receive usual care, which will be monitored. Upper limb therapy will not be restricted for participants during the control wait list phase, rather participants will receive ‘usual care’ conditions [49]. Consistent with independent review and analysis of ‘usual care’ usually received in health services in Australia (based on national audit and knowledge translation study), this will usually not involve specific sensory rehabilitation, such as SENSe therapy [49]. Participants may continue to receive usual care during the SENSe therapy intervention period. Our prior controlled clinical trials provide evidence that exposure to sensory stimuli alone and/or current usual care is not usually sufficient to effect clinically significant improvement in somatosensory function [39, 41, 42]. All services and usual care received will be monitored.

  1. One of the scientific limitations with restoring upper limb movement in stroke is the lack of comparative physiological models and upper limb tasks that delineate loss of functions beyond that known with healthy aging. This is especially critical as there is evidence of nuanced deficits in upper limb sensorimotor functions with normal aging, beyond the conventional gross motor deficits attributable to muscle dynamics. Consequently, the manuscript needs to highlight the loss of sensorimotor function post stroke that is above and beyond that documented in healthy older individuals. Following articles could help authors contextualize this point in the introduction section:
  2. Pagano S, Fait E, Brignani D, Mazza V. Object individuation and compensation in healthy aging. Neurobiol Aging. 2016 Apr;40:145-154. doi: 10.1016/j.neurobiolaging.2016.01.013
  3. Rao N, Mehta N, Patel P, Parikh PJ. Effects of aging on conditional visuomotor learning for grasping and lifting eccentrically weighted objects. J Appl Physiol (1985). 2021 Sep 1;131(3):937-948. doi: 10.1152/japplphysiol.00932.2020
  4. Ingram LA, Butler AA, Lord SR, Gandevia SC. Use of a physiological profile to document upper limb motor impairment in ageing and in neurological conditions. J Physiol. 2023 Jun;601(12):2251-2262. doi: 10.1113/JP283703

Thank you for highlighting this important issue. In the protocol the presence of somatosensory and sensorimotor deficits are identified beyond that known with healthy aging and healthy older individuals, based on age appropriate normative standards. Each of the quantitative somatosensory measures employed use age matched normative standards when defining presence of impairment (Carey et al., 2020; Carey et al., 1996; Mak-Yuen et al., 2023). Moreover, measures used such as the Tactile Discrimination Test and Wrist Position Sense Test have a strong neurobiological pedigree both in their development (e.g. Ben-Shabat, 2008; Carey et al., 2020; Carey et al., 1996, 1997) and in related brain imaging studies (Ben-Shabat et al., 2015; Carey et al., 2008; Carey et al., 2011; Carey et al., 2006; Goodin et al., 2018; Koh et al., 2021). We have also conducted activation likelihood estimation (ALE) meta-analysis of functional neuroimaging studies to contextualise these measures and their activation (Kenzie et al., 2018; Lamp et al., 2018), and have investigated the contribution of somatosensory impairment to pinch deficit and compared with age matched healthy controls (Blennerhassett et al., 2006; Blennerhassett et al., 2008; Blennerhassett et al., 2007).

We have now identified and made more explicit response to this point in the Introduction, as follows: (lines 96 to 97)

Impaired sensation is experienced by one in two stroke survivors [22-25]. This loss is beyond any reduction experienced with healthy aging [22, 23]. 

And noted (lines 120 to 122) that:

Somatosensory impairment and recovery will be quantified using well validated measures with age-matched normative standards and strong psychometric and neuroimaging foundations [34-38]. 

In the data analysis section, it is noted that outcomes will be: ‘…calibrated to normed age-matched performance [39, 50].’

  1. While the authors state that therapists will not be blinded towards the delivery of SENSe therapy to stroke survivors for ‘pragmatic reasons’, it is essential to ensure that the researcher and/or the personnel conducting data processing and analysis of the primary and secondary outcomes is blinded to the assignment of stroke survivors to either group, to reduce the chance of implicit biases.

Thank you. Yes, the personnel conducting data processing and analysis of the primary and secondary outcomes will be blinded to the design to reduce the chance of implicit biases. This detail has now been added to the manuscript, lines 260 to 261.

Researchers and statisticians involved in data analysis will also be blinded to study de-sign and timing of therapy delivery.

  1. Inclusion/exclusion criteria for therapists to be enrolled in the SENSe therapy training need to be standardized to ensure factors including (but not limited to) prior expertise, number of years of practice, number of stroke survivors treated before enrollment to SENSe ‘upskilling’, etc. does not structurally affect the outcomes of the study.

We have now added the details below to better characterise recruitment of therapist participants, as follows: (lines 303 to 306)

As a pragmatic implementation trial, restrictions will not be imposed on therapist participants’ experience or number of stroke survivors treated. However, demographic information for all therapist participants will be collected to ascertain relationships between stroke survivor outcomes and therapist characteristics.

We have also made comment in the Discussion limitations, as below: (lines 617 to 621)

As a pragmatic trial, factors such as individual characteristics of treating therapists (e.g. prior expertise, number of years of practice, number of stroke survivors treated before upskilling) and survivors of stroke (e.g. age, time since stroke, concomitant impairments) will not be controlled for. These factors will however be monitored and explored for their potential influence on outcomes.

  1. It is important to highlight whether the time since stroke will be incorporated as an inclusion criterion for recruitment of stroke survivors. Homogenizing this factor is important because therapists delivering SENSe therapy to stroke survivors within their first 6 months post stroke could observe a greater recovery of sensorimotor function to that observed among survivors having past 12 months or more post stroke. Not restricting this factor could potentially confound the effect and interpretation of SENSe therapy on the outcomes.

As a pragmatic study and based on findings from our prior randomised controlled trial (Carey et al., 2011), we do not wish to restrict recruitment of survivors of stroke to a narrow time post-stroke. Evidence supporting this implementation study indicates that survivors of stroke may benefit from SENSe therapy at varying times post-stroke, ie in subacute and chronic phases (Carey et al., 2011; Carey et al., 2010). 

We previously included the following statements regarding time since stroke in the section of recruitment of stroke survivors (starting line 342).

Stroke survivors: People with stroke living in the community and presenting with new or chronic somatosensory impairment will be recruited….. Inclusion criteria: a clinical diagnosis of stroke (including infarct or hemorrhage, people with a second stroke, no restriction on time since stroke)…. Thus, the design tests effectiveness with relatively unselected survivors of stroke in the community.

We do however acknowledge that time since stroke could be a factor impacting ability to benefit from therapy. Our analysis of therapeutic gain will be benchmarked to that included in our RCT that included participants at a median of 48.14 weeks post stroke with interquartile range of 22.18 to 130.86 weeks (ie wide variance). Further, we will explore individual patient characteristics that impact outcome in this cohort, as we did for the SENSe trial. We have included the following additional text in the data analysis section (lines 494 to 495)

Individual participant characteristics that may impact magnitude of therapeutic gain, such as age and time post-stroke will be explored.

As included in the Discussion: ‘A pragmatic approach was selected to evaluate the effectiveness of our implementation intervention in real-world clinical practice and community settings, and across a range of therapists, to maximize generalizability of the results [68].’ (lines 601 to 603)

We also highlight findings from our previous studies that ‘…A pooled analysis of individual differences indicates adult survivors of stroke with varying age, lesion of left or right hemisphere, severity of impairment, cognition and varying time post-stroke can benefit from SENSe therapy [59]. (lines 586 to 589)

Finally, we have included additional text in the limitations, as below (lines 617 to 621)

As a pragmatic trial, factors such as individual characteristics of treating therapists (e.g. prior expertise, number of years of practice, number of stroke survivors treated before upskilling) and survivors of stroke (e.g. age, time since stroke, concomitant impairments) will not be controlled for. These factors will however be monitored and explored for their potential influence on outcomes.

  1. Why is the Fugl-Myer Upper Extremity test (FM-UE, an established test to evaluate the state of upper extremity function in stroke survivors) not included in one of the outcomes of this protocols?

We acknowledge the FM-UE as a useful upper limb outcome. However, we have not included it in this study due to other outcomes collecting similar data already being used. The outcomes used focus on arm use and include: Jebsen Taylor Hand Function Test (JTHFT) and Motor Activity Log (MAL).

  1. Assignment (random or otherwise) of the stroke survivors to SENSe therapies/'usual core' groups needs to be described in the manuscript. 

The design involves a waitlist control design rather than separate ‘usual group’. Thus all stroke survivors receive a phase of usual care and a phase of SENSe therapy. The design is reported in lines 202 to 203 and 233 to 238.

It should be noted that the focus of the current study is on implementation rather than treatment effectiveness (which was investigated in the prior RCT (Carey et al., 2011). Here the focus is on delivery of the therapy and benchmarking effectiveness relative to therapeutic gain described in the RCT.

Comments on the Quality of English Language

Proof-reading could help fix occasional typos in the text.

Thank you we have conducted proof-reading of the manuscript and have made minor changes accordingly. Changes are highlighted in the revised manuscript.

Additional references linked to Reviewer 3 responses above:

Ben-Shabat, E. (2008). Central processing of proprioception: Functional MRI and psychophysical studies in healthy and stroke participants LaTrobe University].

Ben-Shabat, E., Matyas, T. A., Pell, G. S., Brodtmann, A., & Carey, L. M. (2015). The right supramarginal gyrus is important for proprioception in healthy and stroke-affected participants: A functional MRI study. Frontiers in Neurology, 6. https://doi.org/10.3389/fneur.2015.00248

Blennerhassett, J. M., Carey, L. M., & Matyas, T. A. (2006). Grip force regulation during pinch grip lifts under somatosensory guidance: comparison between people with stroke and healthy controls. Archives of Physical Medicine and Rehabilitation, 87(3), 418-429. http://www.ncbi.nlm.nih.gov/entrez/query.fcgi?cmd=Retrieve&db=PubMed&dopt=Citation&list_uids=16500179

Blennerhassett, J. M., Carey, L. M., & Matyas, T. A. (2008). Impaired discrimination of sensory information about slip between object and skin is associated with handgrip limitation poststroke. Brain Impairment, 9(2), 114-121. https://doi.org/10.1197/j.jht.2007.10.021

Blennerhassett, J. M., Matyas, T. A., & Carey, L. M. (2007). Impaired discrimination of surface friction contributes to pinch grip deficit after stroke. Neurorehabilitation and Neural Repair, 21(3), 263-272. https://doi.org/10.1177/1545968306295560

Cahill, L. S., Lannin, N. A., Mak-Yuen, Y. Y., Turville, M. L., & Carey, L. M. (2018). Changing practice in the assessment and treatment of somatosensory loss in stroke survivors: Protocol for a knowledge translation study. BMC Health Services Research, 18(1), 1-8.

Cahill, L. S., Lannin, N. A., Purvis, T., Cadilhac, D. A., Mak-Yuen, Y., O'Connor, D. A., & Carey, L. M. (2022). What is "usual care" in the rehabilitation of upper limb sensory loss after stroke? Results from a national audit and knowledge translation study. Disabil Rehabil, 44(21), 6462-6470. https://doi.org/10.1080/09638288.2021.1964620

Carey, L., Macdonell, R., & Matyas, T. A. (2011). SENSe: Study of the Effectiveness of Neurorehabilitation on Sensation: A randomized controlled trial. Neurorehabilitation and Neural Repair, 25(4), 304-313. https://doi.org/10.1177/1545968310397705

Carey, L. M., Abbott, D. F., Egan, G. F., & Donnan, G. A. (2008). Reproducible activation in BA2, 1 and 3b associated with texture discrimination in healthy volunteers over time. Neuroimage, 39(1), 40-51. http://www.ncbi.nlm.nih.gov/entrez/query.fcgi?cmd=Retrieve&db=PubMed&dopt=Citation&list_uids=17911031

Carey, L. M., Abbott, D. F., Harvey, M. R., Puce, A., Seitz, R. J., & Donnan, G. A. (2011). Relationship between touch impairment and brain activation after lesions of subcortical and cortical somatosensory regions. Neurorehabilitation and Neural Repair, 25(5), 443-457. https://doi.org/10.1177/1545968310395777

Carey, L. M., Abbott, D. F., Puce, A., Seitz, R. J., Harvey, M., & Donnan, G. A. (2006). IN_Touch: Imaging Neuroplasticity of Touch post-stroke with moderate and severe touch impairment. Neuroimage, 31(Suppl 1), e1.

Carey, L. M., Mak-Yuen, Y. Y. K., & Matyas, T. A. (2020). The functional tactile object recognition test: A unidimensional measure with excellent internal consistency for haptic sensing of real objects after stroke. Frontiers in Neuroscience, 14(948). https://doi.org/10.3389/fnins.2020.542590

Carey, L. M., Matyas, T., Walker, J., & Macdonell, R. (2010). SENSe: Study of the Effectiveness of Neurorehabilitation on Sensation: Individual patient characteristics that predict favourable outcomes 21st Annual Scientific Meeting of the Stroke Society of Australasia, Melbourne, Australia.

Carey, L. M., Oke, L. E., & Matyas, T. A. (1996). Impaired limb position sense after stroke: a quantitative test for clinical use. Archives of Physical Medicine and Rehabilitation, 77(12), 1271-1278.

Carey, L. M., Oke, L. E., & Matyas, T. A. (1997). Impaired touch discrimination after stroke: a quantitative test. Neurorehabilitation and Neural Repair, 11(4), 219-232.

Goodin, P., Lamp, G., Vidyasagar, R., McArdle, D., Seitz, R. J., & Carey, L. M. (2018). Altered functional connectivity differs in stroke survivors with impaired touch sensation following left and right hemisphere lesions. Neuroimage Clinical, 18, 342-355. https://doi.org/10.1016/j.nicl.2018.02.012

Kenzie, J. M., Ben-Shabat, E., Lamp, G., Dukelow, S. P., & Carey, L. M. (2018). Illusory limb movements activate different brain networks than imposed limb movements: an ALE meta-analysis. Brain Imaging Behav, 12(4), 919-930. https://doi.org/10.1007/s11682-017-9756-1

Koh, C. L., Yeh, C. H., Liang, X., Vidyasagar, R., Seitz, R. J., Nilsson, M., Connelly, A., & Carey, L. M. (2021). Structural connectivity remote from lesions correlates with somatosensory outcome poststroke. Stroke, 52(9), 2910-2920. https://doi.org/10.1161/strokeaha.120.031520

Lamp, G., Goodin, P., Palmer, S., Low, E., Barutchu, A., & Carey, L. M. (2018). Activation of Bilateral Secondary Somatosensory Cortex With Right Hand Touch Stimulation: A Meta-Analysis of Functional Neuroimaging Studies. Front Neurol, 9, 1129. https://doi.org/10.3389/fneur.2018.01129

Mak-Yuen, Y. Y. K., Matyas, T. A., & Carey, L. M. (2023). Characterizing touch discrimination impairment from pooled stroke samples using the Tactile Discrimination Test: Updated criteria for interpretation and brief test version for use in clinical practice settings. Brain Sciences, 13(4). https://doi.org/10.3390/brainsci13040533 

Reviewer 4 Report

Comments and Suggestions for Authors

It is honor to review your manuscript. 

Line 78: It was said that Evidence-based therapies for the upper limb after stroke are recommended in clinical practice guidelines, but there was only one reference.

Please provide a figure or detailed explanation of SENSe therapy.

Please mention limitation.

Author Response

Thank you. Please find response to reviewer comments below, with new text indicated in red. The final revised manuscript with changes tracked is also attached below.

It is honor to review your manuscript. 

Line 78: It was said that Evidence-based therapies for the upper limb after stroke are recommended in clinical practice guidelines, but there was only one reference.

Thank you. This has now been corrected, as below. The two references to country guidelines [3,4] are now included here, and refs [5,6] used for the best practice and synthesis references, refer to multiple guidelines. Please also note that the plural ‘guidelines’ was originally used, consistent with the reference to the Australian guidelines (as below). 

Evidence-based therapies for the upper limb after stroke are recommended in clinical practice guidelines [3, 4] and in international best-practice guidelines [5, 6].

  1. Stroke Foundation, Clinical guidelines for stroke management 2022 (living guidelines). 2022: Stroke Foundation https://informme.org.au/en/Guidelines/Clinical-Guidelines-for-Stroke-Management.
  2. Intercollegiate Stroke Working Party. National clinical guideline for stroke for the United Kingdom and Ireland. 2023, Royal College of Physicians: London, UK DOI: https://www.strokeguideline.org/contents/.
  3. Wolf, S.L.; Kwakkel, G.; Bayley, M.; McDonnell, M.N.; for the Upper Extremity Stroke Algorithm Working Group. Best practice for arm recovery post stroke: An international application. Physiotherapy, 2016. 102(1), 1-4 DOI: http://dx.doi.org/10.1016/j.physio.2015.08.007.
  4. Mead, G.E.; Sposato, L.A.; Sampaio Silva, G.; Yperzeele, L.; Wu, S.; Kutlubaev, M.; Cheyne, J.; Wahab, K.; Urrutia, V.C.; Sharma, V.K.; Sylaja, P.N.; Hill, K.; Steiner, T.; Liebeskind, D.S.; Rabinstein, A.A. A systematic review and synthesis of global stroke guidelines on behalf of the world stroke organization. International Journal of Stroke, 2023. 18(5), 499-531 DOI: 10.1177/17474930231156753.

Please provide a figure or detailed explanation of SENSe therapy.

Thank you. During SENSe therapy seven training principles, based on neuroscience and learning, are used during the process of somatosensory discrimination training to drive change (http://youtu.be/G9V3I30pn68). In addition, specialised training equipment is used. We have now included a more detailed explanation of SENSe therapy as below in red, at lines 436 to 447.

SENSe therapy involves modules of sensory discrimination training of texture discrimination, sensing the position of the upper limb in space (proprioception), recognizing objects through the sense of touch, and learning how to apply these skills in daily tasks identified by the person with stroke. Seven training principles are used during the process of somatosensory discrimination training as follows: select; attentive exploration: feedback; calibrate; anticipate; repeat and progress; transfer (http://youtu.be/G9V3I30pn68). Specially designed training tools are used and include: grids and texture training wheels of surfaces with varying surface features, graded for large, medium and fine surface differences; box-like apparatus and protractor scales for training wrist position sense; and graded sets of objects for functional tactile object recognition that train discrimination and recognition of object weight, crushability, shape, size, temperature, texture, and functional motion. The client selects two valued activities that they believe are impacted by their sensory loss to focus on in therapy. They are guided to discover the sensory challenges in the activity and how they can use their new skills to perform the task better. Examples include, using a knife or fork, finding money in wallet, doing up buttons, using a remote-control device. The intervention will be tailored according to level of impairment and functional goals of the stroke participant. The dose is 10 one-hour sessions, over a period of 6 weeks. The frequency of sessions is approximately twice a week. The number, duration and specific content of all sessions is monitored using customized training forms.

We also provide a figure to capture some of the relevant training modules and application to activities in SENSe therapy, see new Figure 2, in revised manuscript attached (I am unable to include here).

Figure 2: Images of SENSe therapy specialized equipment and activities used in training modules, including a) discrimination of texture grids; b) training of wrist proprioception using the box-like apparatus and protractor scales; c) training of graded sets of everyday objects that vary in objects with varying diagnostic attributes such as crushability, shape, size, weight; d) training in context of self-chosen valued activities impacted by sensory loss.

Please mention limitation.

As a protocol paper, limitations will be fully described in results publications. However, we have also included a brief paragraph on limitations in the Discussion, as below (lines 620 to 630).

Limitations: Increased access to delivery of evidence-based SENSe therapy via specialist SENSe therapy centers, the primary outcome, will likely be impacted by restrictions imposed during the COVID pandemic. The ability to administer the protocol as planned and deliver SENSe therapy without interruption will also likely be impacted. Changed service demands associated with the pandemic may also influence recruitment and retention of therapists and delivery of usual care. Any protocol variations and impacts of the pandemic will be monitored and reported. As a pragmatic trial, factors such as individual characteristics of treating therapists (e.g. prior expertise, number of years of practice, number of stroke survivors treated before upskilling) and survivors of stroke (e.g. age, time since stroke, concomitant impairments) will not be controlled for. These factors will however be monitored and explored for their potential influence on outcomes.

Reviewer 5 Report

Comments and Suggestions for Authors

The manuscript entitled “  A network of sites and upskilled therapists to deliver 2 best-practice stroke rehabilitation of the arm: Protocol for 3 a Knowledge Translation Study’: is well written and very organized and the authors cooperated well to establish their manuscript but I have a major concern regarding the way they presented their results , no histograms or tables about the obtained statistical data and no fighures for any of successful trials for their prospective analysis.

-        I found the manuscript is well presented but the key out comes are not well defined. Please , revise.

-        Line 87 : “ Theoretical Domains Framework” does this has abbreviation.

-        What do you mean by “ Reduced sensation is experienced by one in two stroke survivors”

-        Line 96 : is here a missed reference ““…I couldn’t really do 96 daily stuff [...]”?

-        Re-104 habilitation therapists often use a compensatory, rather than restorative, approach to 105 somatosensory loss and recovery [17, 18]; potentially reinforcing learned non-use of the 106 limb with negative long-term consequences.” This part seems not well presented and the references should be transferred to the end of the sentence because it interrupt the meaning, Please rephrase.

-        What you mean by “ signaling a readiness 109 to change practice” at the end of line 109.

-        The last point in the aim of the work needs paraphrasing and revision.

-        What is this reference refer to “ This knowledge translation study and implementation intervention developed by us  [12],” , what does this sentence mean ?

-        In this section “ Methods to facilitate sustainability” all is expressing about what you are aiming to achieve in the future, So it is more suitable in the recommendation section. How you feel?

-         

Comments on the Quality of English Language

Moderate English Editing is needed

Author Response

Thank you. Please find below response to the reviewer's comments, with new text highlighted in red. The final revised manuscript with tracked changes is also attached.

The manuscript entitled “  A network of sites and upskilled therapists to deliver best-practice stroke rehabilitation of the arm: Protocol for a Knowledge Translation Study’: is well written and very organized and the authors cooperated well to establish their manuscript but I have a major concern regarding the way they presented their results , no histograms or tables about the obtained statistical data and no fighures for any of successful trials for their prospective analysis.

It is important to emphasise that this is a protocol paper. No results are presented.

I found the manuscript is well presented but the key out comes are not well defined. Please , revise.

The outcomes are described in the text from lines 335 and are listed in Table 1. At both points in the manuscript, full references are also cited for each of the outcomes.

We have now provided additional text for the key outcomes, as below:

 Stroke survivors: The primary outcome is change in arm somatosensory function, pre-post SENSe therapy, across three somatosensory domains, measured using standardized, quantitative somatosensory measures and a normalized summary impairment index calibrated to normed age-matched performance [39, 50]. The index will be derived from scores on the Tactile Discrimination Test [38], Wrist Position Sense Test [35], and functional Tactile Object Recognition Test [37]. Each of these quantitative measures assesses the person’s ability to discriminate different somatosensations; i.e. texture discrimination, wrist joint proprioception, and haptic object recognition of the upper limb. Comparable ranges of impairment from just noticeable to extreme impairment defined for each measure enable the normalization of the three test scales for comparison in clinical and research settings [50].

Line 87 : “ Theoretical Domains Framework” does this has abbreviation.

Yes, TDF is used as an abbreviation for Theoretical Domains Framework. However, it is only mentioned once in manuscript and therefore does not need a recurrent abbreviation.

What do you mean by “ Reduced sensation is experienced by one in two stroke survivors”

This means that approximately 50% of survivors of stroke experience impaired somatosensation after stroke. We have chosen to report this frequency at the person level by stating ‘one in two stroke survivors’. We have updated this text for clarity and in response to Reviewer 3, as below. (line 96-97)

Impaired sensation is experienced by one in two stroke survivors [22-25]. This loss is beyond any reduction experienced with healthy aging [22, 23].

Line 96 : is here a missed reference ““…I couldn’t really do daily stuff [...]”?

Thank you for identifying this error. The appropriate reference [26] is provided at the end of the quotes. The brackets have now been deleted at this mid point.

“ Rehabilitation therapists often use a compensatory, rather than restorative, approach to somatosensory loss and recovery [17, 18]; potentially reinforcing learned non-use of the limb with negative long-term consequences.” This part seems not well presented and the references should be transferred to the end of the sentence because it interrupt the meaning, Please rephrase.

Text has been rephased as below. (lines 106 to 109)

Rehabilitation therapists often use a compensatory, rather than restorative, approach to somatosensory loss and recovery [17, 18]. Yet, use of compensation potentially reinforces learned non-use of the limb, with negative long-term consequences.

What you mean by “ signaling a readiness to change practice” at the end of line 109.

We are unsure why this unclear. However, have now used the word ‘indicating’ instead of ‘signaling’ as this is likely in more common use. (line 112)

The last point in the aim of the work needs paraphrasing and revision.

We have rephrased as below. (see line 194 to 196)

Evaluate the cost-effectiveness of the knowledge translation intervention in terms of the amount of improvement in SENSe therapy outcomes, i.e. somatosensory function, performance in valued activities, arm use, and quality of life

What is this reference refer to “ This knowledge translation study and implementation intervention developed by us  [12],” , what does this sentence mean ?

The reference refers to our published study where the implementation intervention developed by the research team to change therapist behavior has been described in detail, eg including the multimodal approach to upskilling therapists, as part of the SENSe Implement study (Cahill et al., 2018). As the sentence you highlight is broader than this and refers to the current protocol, we now make clear that we are referring to the current study and research team and have deleted the reference to our earlier publication, as below. We trust that this clarifies our meaning. (lines 397 to 400)

The current knowledge translation study and implementation intervention developed by the research team, are based on the Theoretical Domains Framework [14], and guided by strategies from the Behaviour Change Wheel [16] to facilitate practice change, and Normalisation Process Theory [56] to enhance sustainability.

In this section “ Methods to facilitate sustainability” all is expressing about what you are aiming to achieve in the future, So it is more suitable in the recommendation section. How you feel?

Sustainability is a key consideration for knowledge translation and should remain in the Methods section, together with the implementation intervention. As it is part of our methods that is pre-planned and employed from the outset of the study, we strongly argue it is important retain in our protocol methods.

Comments on the Quality of English Language: Moderate English Editing is needed

Thank you we have conducted proof-reading of the manuscript and have made changes accordingly and in response to reviewers’ comments and suggestions. Changes are highlighted in the revised manuscript.

Additional reference included in response to Reviewer 5 above.

Cahill, L. S., Lannin, N. A., Mak-Yuen, Y. Y., Turville, M. L., & Carey, L. M. (2018). Changing practice in the assessment and treatment of somatosensory loss in stroke survivors: Protocol for a knowledge translation study. BMC Health Services Research, 18(1), 1-8.

Reviewer 6 Report

Comments and Suggestions for Authors

As this reviewer understands this work it is a protocol for a multi center study to test the effectiveness of their proposed treatment called SENSe.

More likely it is to be submitted to the IRB at their institutions or to the New England Journal for prepublication of the study plan so that the results can sluice through the system quickly - especially if the results are positive. 

This reviewer does not have time to go to the referenced papers which presumably focus on the effectiveness of the SENSe treatment protocol, however there is agreement that preserved sensation in the affected upper limb- especially proprioception - may well predict future motor outcome better than current descriptions of finger-thumb opposition and on and on. In that light a focus on sensory recovery may well usher in new ideas about restorative treatment modalities.

Author Response

Thank you. Please find below response to Reviewer 6 comments. The final revised manuscript with tracked changes is also attached.

As this reviewer understands this work it is a protocol for a multi center study to test the effectiveness of their proposed treatment called SENSe.

More likely it is to be submitted to the IRB at their institutions or to the New England Journal for prepublication of the study plan so that the results can sluice through the system quickly - especially if the results are positive. 

Yes, as indicated, this study is a protocol for a multicenter study. The focus is on implementation of an evidence-based therapy known as SENSe rather than an effectiveness study of SENSe (this has been published in our randomised controlled trial and other publications). The protocol has been approved by central and institutional ethics committees. As indicated, the intent is to prepublish the protocol in readiness for reporting of results. We have submitted to Healthcare, where we believe the readership will be interested in this implementation protocol that involves a network of sites and upskilled therapists and details an application of knowledge translation for complex interventions that is virtually untested in the field of stroke rehabilitation [12].

This reviewer does not have time to go to the referenced papers which presumably focus on the effectiveness of the SENSe treatment protocol, however there is agreement that preserved sensation in the affected upper limb- especially proprioception - may well predict future motor outcome better than current descriptions of finger-thumb opposition and on and on. In that light a focus on sensory recovery may well usher in new ideas about restorative treatment modalities.

Thank you. We agree with the reviewer’s view that ‘…a focus on sensory recovery may well usher in new ideas about restorative treatment modalities.’ In line with this view, we further highlight the importance of a multicentre study to facilitate the implementation of this evidence-based intervention.

Round 2

Reviewer 3 Report

Comments and Suggestions for Authors

The authors do a commendable job at addressing earlier comments. The concern with presently incorporated revision to one of the comments is highlighted below:

1.     The response to reviewer’s earlier comments on a rather focal issue on loss of function in stroke above and beyond aging (labeled 3-6 in authors’ reply) seems divergent and elongated. Specifically, the authors include two additions noted below:

“… Impaired sensation is experienced by one in two stroke survivors [22-25]. This loss is beyond any reduction experienced with healthy aging [22, 23]…”

And

“Somatosensory impairment and recovery will be quantified using well validated measures with age-matched normative standards and strong psychometric and neuroimaging foundations [34-38].”

While the authors’ self-citation is justified for their first addition, their second addition (and associated self-citation) is unnecessary. The removal of second correction is essential to avoid confusion for the readers.

2.     This comment is just to bring to the authors’ attention if they would like to deidentify (by blurring, blocking, or via other method) the person depicted in Figure 2.

Author Response

The authors do a commendable job at addressing earlier comments. The concern with presently incorporated revision to one of the comments is highlighted below:

  1. The response to reviewer’s earlier comments on a rather focal issue on loss of function in stroke above and beyond aging (labeled 3-6 in authors’ reply) seems divergent and elongated. Specifically, the authors include two additions noted below:

“… Impaired sensation is experienced by one in two stroke survivors [22-25]. This loss is beyond any reduction experienced with healthy aging [22, 23]…”

And

“Somatosensory impairment and recovery will be quantified using well validated measures with age-matched normative standards and strong psychometric and neuroimaging foundations [34-38].”

While the authors’ self-citation is justified for their first addition, their second addition (and associated self-citation) is unnecessary. The removal of second correction is essential to avoid confusion for the readers.

Thank you. We have now removed the second correction and associated references to avoid confusion. Text has been deleted from line 121. Previous references 34 to 38 have been deleted at this point and reference citations updated.

  1. This comment is just to bring to the authors’ attention if they would like to deidentify (by blurring, blocking, or via other method) the person depicted in Figure 2.

Thank you. We have updated the figure accordingly.

Please also find attached updated manuscript with revision 2 updates highlighted in yellow.

Reviewer 5 Report

Comments and Suggestions for Authors

I found the authors worked well to improve their manuscript and addressed the required points qdequately. 

Eventhough, I would appreciate if they can add the limitations of the data collection and analysis and specialists training. 

Comments on the Quality of English Language

Minor English Editing is required 

Author Response

I found the authors worked well to improve their manuscript and addressed the required points adequately. 

Eventhough, I would appreciate if they can add the limitations of the data collection and analysis and specialists training. 

Thank you. These are now added to the Limitations section of the Discussion, as below in red (see lines 600 to 611).

As a pragmatic trial, factors such as individual characteristics of treating therapists (e.g. prior expertise, number of years of practice, number of stroke survivors treated before upskilling) and survivors of stroke (e.g. age, time since stroke, concomitant impairments) will not be controlled for. These factors will however be monitored and explored for their potential influence on outcomes in our planned analyses. Therapists who deliver usual care will be different from those who deliver SENSe therapy. It is recognized that this may impact outcomes. It is noted however that involvement of different therapists in this pragmatic design is consistent with variation in therapist experience typically experienced. Further, it is not possible for the SENSe therapists to deliver usual care following upskilling, as the specialist training may bias their usual care approach. Usual care will not be restricted, nor will it involve standard protocols. Rather it will be monitored and compared to usual care as defined in our aligned SENSe Implement study, that included a phase of delivery of usual care (n= 86 patients) before therapists were upskilled in existing healthcare services, and with independent analysis of the care usually received in health services based on a national audit [44]. To date there are no results to present for the current prospective study.

Comments on the Quality of English Language

Minor English Editing is required 

Text has been closely reviewed, edited, and updated, as indicated in the revised manuscript.

Please find attached updated manuscript with revision 2 updates highlighted in yellow.
